# Review of Jackson Herring's Early Work on Thermal Convection

**Robert M. Kerr**

Department of Mathematics, University of Warwick, Coventry CV4 7AL, UK; robert.kerr@warwick.ac.uk

**Abstract:** Jack Herring had three mid-1960s numerical papers on Rayleigh-Bénard thermal convection that might seem primitive by today's standards, but already encapsulated many of the questions that are still being asked. All of them use severely truncated versions of the incompressible Navier–Stokes–Boussinesq equations with only one, or just a few, horizontal Fourier modes. In the first two papers, 1963 and 1964, the presented results used only one Fourier mode $\alpha$ and three variables. The single mode's variables are its vertical velocity profile $w_\alpha(z,t)$, its temperature profile $\theta_\alpha(z,t)$ and the horizontally uniform vertical profile of the background temperature $\psi(z,t)$. All of the second- and third-order terms are ignored except the convective heat flux $\overline{w\theta}$. The objective was to find asymptotic steady-state solutions. Each paper found evidence for the one-third Nusselt versus Rayleigh scaling of $Nu \sim Ra^{1/3}$, originally derived from Malkus' maximum flux principle. The 1963 paper uses free-slip upper and lower boundaries, with magnitudes of $Nu$ that are a factor of three larger than the experiments. In the 1964 paper, by introducing no-slip/rigid boundary conditions, the magnitude of $Nu$ dropped to within 20% of the experimental values. Both $Nu(Ra)$ relations are in good agreement with circa-1990 direct numerical simulations (DNS). This dependence upon the boundary condition at the walls suggests that to obtain physically realistic scaling, no-slip boundary conditions are necessary. The third paper is discussed only in terms of what it might have been aiming to accomplish and its relation to the earlier free-slip results.

**Keywords:** convection; turbulence; scaling laws





## 1. Introduction

When Jack Herring's mid-1960s convection papers [1–3] were published, there were already, as there are now, two primary predictions for the dependence of the dimensionless heat flux, the Nusselt number, $Nu$, upon the Rayleigh number, $Ra$. From Malkus [4,5] and Kraichnan [6], a scaling law for modest Rayleigh numbers was derived; from Kraichnan, an upper bound on the scaling for asymptotically large $Ra$ was derived, suggesting, respectively,

$$Nu \sim CRa^{1/3} \quad \text{and} \quad Nu \sim CRa^{1/2}/(\log Ra)^{3/2} \quad \text{with} \quad Ra \equiv \frac{\alpha_e g \Delta T d^3}{\kappa \nu}. \tag{1}$$

where the variables are as follows: $\alpha_e$ is the linear coefficient of expansion; $g$ is the gravitational acceleration; $\Delta T = (T_H - T_C)$ is the temperature difference between the hot and cold walls; $d$ is the depth; $\kappa$ is the thermal conductivity; and $\nu$ is the kinematic viscosity. A $Nu \sim CRa^{1/2}$ scaling law had also been suggested by Spiegel [7].

To obtain some numerical insight into whether either prediction might hold, Herring used very elementary numerics that incorporated the primary assumptions described by Malkus and Veronis [5]. Using a few horizontal Fourier modes at most, the variables of the reduced dynamics were the vertical profiles of the poloidal vertical velocities and temperatures. Velocities and the vertical vorticity from the toroidal component [8] were neglected. The goal was to find steady-state solutions by iterating in time. For the primary calculations, only one Fourier mode was used and, in each case, the $Nu \sim CRa^{1/3}$ scaling prediction was supported, with geometric and Reynolds-number-dependent coefficients,

*C*, in approximate agreement with 1990s-era direct numerical simulations (DNS) for both free(-slip) [9] and no-slip (rigid) boundaries [10]. No evidence was found for an ultra-high Rayleigh number $Nu \sim Ra^{1/2}$ regime.

This review begins by describing the reduced equations and the iteration method and then the results. First the free-slip calculations will be discussed, and then we will move on to those with rigid boundaries. The 1966 paper is not discussed in any detail because the 1963 scaling results are repeated. Then there is a discussion of energy budgets, the relationships to the first DNS direct numerical simulations and the overall message.

## 2. Governing Equations

The full incompressible Navier–Stokes–Boussinesq equations for the velocity vector $\boldsymbol{v}(\boldsymbol{r}, t)$ and temperature fields $T(\boldsymbol{r}, t)$ for Prandtl number $\sigma = \nu / \kappa$ are:

$$\nabla \cdot \boldsymbol{v} = 0 , \tag{2}$$

$$\left( \frac{1}{\sigma} \frac{\partial}{\partial t} - \nabla^2 \right) \nabla^2 \boldsymbol{v} = \frac{1}{\sigma} \nabla \times \nabla \times (\boldsymbol{v} \cdot \nabla \boldsymbol{v}) + Ra \nabla \times (\nabla \times \boldsymbol{k} T) , \tag{3}$$

$$\left( \frac{\partial}{\partial t} - \nabla^2 \right) T = -\nabla \cdot (\boldsymbol{v} T) . \tag{4}$$

From these equations, the dimensional variables (denoted by primes) are made non-dimensional using the depth $d$ of the convection cell and $\kappa$.

$$\boldsymbol{v} = \frac{d}{\kappa} \boldsymbol{v}', \qquad T = T'/d,$$

$$\boldsymbol{r} = \boldsymbol{r}'/d, \qquad t = \frac{\kappa}{d^2} t' .$$

The double curl has eliminated the pressure from the equations and by considering only the poloidal component, one is left with only one velocity component: the vertical velocity $w(\boldsymbol{r}, t)$. The temperature field $T(\boldsymbol{r}, t)$ is then separated into a fluctuating part $\theta$ and a horizontally uniform background field $\psi(z, t)$.

$$T(\boldsymbol{r}, t) = -z + \psi(z, t) + \theta(\boldsymbol{r}, t) \quad \text{with} \quad \beta(z) = -\frac{\partial}{\partial z} \overline{T}(z) = 1 - \frac{\partial \psi}{\partial z} , \tag{5}$$

where $\overline{T}(z)$ is the horizontally averaged temperature. The boundary conditions at $z = 0$ and $z = 1$ on the temperature components are

$$\psi(0, t) = \psi(1, t) = 0 \tag{6}$$

$$\theta(x, y, 0, t) = \theta(x, y, 1, t) = 0. \tag{7}$$

Upon horizontally averaging the equations, and removing all except one of the horizontal advection and second-order fluctuation terms, the system can be reduced further, with the exception being the vertical heat transport $\overline{w\theta}$ that is forcing the horizontally averaged temperature $\psi$ and its derivative $\beta$.

$$\left( \frac{1}{\sigma} \frac{\partial}{\partial t} - \nabla^2 \right) \nabla^2 w_\alpha = Ra \nabla_\perp^2 \theta_\alpha , \tag{8}$$

$$\left( \frac{1}{\sigma} \frac{\partial}{\partial t} - \nabla^2 \right) \theta_\alpha = \beta w , \tag{9}$$

$$\left( \frac{\partial}{\partial t} - \frac{\partial^2}{\partial z^2} \right) \beta = + \frac{\partial^2}{\partial z^2} \overline{w\theta} . \tag{10}$$

where $\nabla_\perp = \partial_x^2 + \partial_y^2$. This is the later 1964 version of the reduced Equations [2].

The horizontally fluctuating terms are then decomposed into horizontal Fourier modes $w(\alpha_j, z, t)$ and $\theta(\alpha_j, z, t)$, where $\alpha_j$ represents a horizontal wavenumber $(k_x, k_y)_j$.

There are two nonlinear terms. $\beta w$ in (9) and $\overline{w\theta}$ in (10). Today, these terms would be found in physical space, transformed to Fourier space using an FFT, derivatives taken, then transformed back to $z$-profiles. These calculations were pre-FFT, so the derivatives in the nonlinear terms are included using convolution—that is, summing in Fourier space the nonlinear, multiplied terms.

Once the time derivatives are set, time-iteration using simple forwards Euler is applied to determine if a steady state can be reached. If a steady state is obtained, this question can be posed: Are the steady-state solutions for the two geometries, free-slip and rigid, consistent with our current understanding, numerical or experimental, of convection in those geometries?

## 3. Herring 1963 Results

In a 1963 paper [1], only one horizontal Fourier mode was used in each simulation and the vertical profiles of all the variables expanded as sums of vertical sine functions. This imposes free-slip boundaries at $z = 0$ and 1 and makes the convolution sums for the nonlinear terms particularly easy.

Choosing values of $\alpha$ and the Rayleigh number that would support convection and not be unstable was not trivial. One restriction due to marginal stability [8] is that the system will not support convection if

$$\frac{(1 + \alpha^2)^3}{\alpha^2} \geq \frac{Ra}{\pi^4} \,. \tag{11}$$

In Section 5 of Ref. [1], the stability is summarized with its Figure 17, including an observation that a small $\alpha$ is enormously unstable. The figures below come from the stable range of $\alpha$ and $Ra$.

As $Ra$ increases, the variations in the boundary layers become stronger, with profiles that are qualitatively similar to those from free-slip DNS [9]. Furthermore, the fit for $Nu(Ra) = C\,Ra^{1/3}$ (1) gives $C = 0.31$, which with some digging, is exactly what the free-slip DNS finds for $Ra = 5 \times 10^5$ with $Nu = 23.7$ [9].

## 4. Herring 1964 Results

In the 1964 paper, rigid, no-slip boundary conditions were applied [2].

However, applying the sine function algorithms used for the free-slip calculations directly to the rigid case is impossible because at each wall the vertical velocity has two boundary conditions, not one.

$$w(0, t) = \frac{\partial w}{\partial z}(0, t) = w(1, t) = \frac{\partial w}{\partial z}(1, t) = 0 \,. \tag{12}$$

To handle this, in the appendix of [2], inhomogeneous combinations of sinh and cosh functions [8] of $\pi\alpha z$ are added to terms based upon sines. The inhomogeneous parts allow all four boundary conditions in (12) to be satisfied. Then, Green's functions relating those inhomogeneous plus homogeneous functions to sines are computed.

With those Green functions, a time-advancing iteration procedure analogous to Equations (8)–(10) can be implemented.

Furthermore, to make things simpler, the Prandtl number limit $\sigma \to \infty$, $\sigma^{-1} \to 0$ was taken, hoping that this would include the $\sigma \sim 1$ of the air around us, an assumption that is partially borne out by the DNS of Kerr and Herring [11].

## 5. Discussion

In the first paper [1], it was shown how a single horizontal Fourier mode with simple parameterizations could generate the proposed $Nu \sim Ra^{1/3}$ scaling [4] for the heat flux. In the second paper [2], by imposing no-slip boundary conditions on the velocity, instead of

free-slip, it was shown how two results could be generated and this was consistent with the observations: the coefficient on the $Ra^{1/3}$ scaling and the vertical profiles of $w$ and $\theta$.

However, this unanswered remains question: Does the asymptotic Kraichnan [6] upper bound for high Rayleigh number scaling, $Nu \sim Ra^{1/2}$, have any relevance? Today, this hypothetical regime is called the ultimate regime.

The third paper [3] was an attempt to pre-condition and accelerate the iteration towards a steady state in order to determine whether there might be additional scaling laws at much higher Rayleigh numbers, as suggested by that upper bound. The pre-conditioning was only applied to the free-slip geometry and there was no evidence for any new regimes as the earlier free-slip results [1] were reproduced.

Another concept being introduced to numerics was the role of what we now call the energy budget [1]. For Rayleigh-Bénard convection, there are two energy functions,

$$E_K = \tfrac{1}{2}\{w^2\}_v \quad \text{and} \quad E_P = \tfrac{1}{2}\left\{|\theta|^2 + |\psi|^2\right\}_v, \tag{13}$$

the kinetic energy $E_K$ [5] and the potential energy $E_P$, which is called entropy in the paper. The $v$-subscripts indicate integration over the volume and the total energy is $E_P + E_K$. The $E_P$ equation is

$$\frac{\partial}{\partial t} E_P + \left\{|\nabla\theta|^2 + |\nabla\psi|^2\right\}_v = \{w\theta\}_v. \tag{14}$$

The energy flow in this reduced system of just $w_\alpha$, $\theta_\alpha$ and $\beta$ is as follows:

- The inflow begins as the vertical velocity $w$ interacts with the mean temperature gradient $\beta$ in (9), whose volume integral is $\{\beta w\}_v$.
- $E_P$ then flows to the kinetic energy through the $\nabla_\perp^{-2} Ra \nabla_\perp^2 \theta$ term in (8).
- This is then dissipatively removed by $\nabla_\perp^{-2}$ inverting the $\nabla_\perp^4$ term of (8):

$$\nabla_\perp^{-2}\left\{\nabla_\perp^2 \nabla_\perp^2 w\right\}.$$

- Meanwhile, $\beta$ is modified by (10).

**Similarities and differences.** Figures 1 and 2 for no-slip boundaries have the following similarities with, and differences from, the free-slip results in Figures 3 and 4,

- $\overline{T}(z)$ and $\beta(z)$ profiles are similar. Except at the walls where $\partial\beta/\partial z \sim 0$.
- Similarly for $w(z)$ and $\theta(z)$, except again $\partial w/\partial z = 0$ at the walls.
- **For no-slip (rigid) walls** the coefficient in front of the $Ra^{1/3}$ given in the abstract is $C = 0.1153$ and is within 20% of the experiments. And consistent with $Ra < 10^6$ DNS [10] for which $C \approx 0.09 \sim 0.11$ if strictly $Nu \sim Ra^{1/3}$ scaling is assumed.

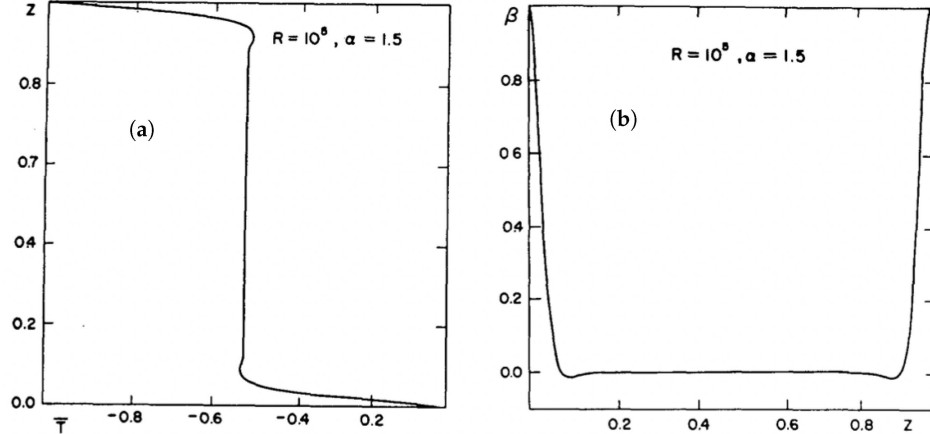

**Figure 1.** Rigid (no-slip) boundaries with $Ra = 10^5$ and $\pi\alpha = 5$. (**a**) Mean temperature $\overline{T}(z)$. (**b**) Mean gradient $\beta(z)$. $\beta(z)$ has been normalized by the total heat transport $Nu = 6.7$. Figure 6a,b from Ref. [2].

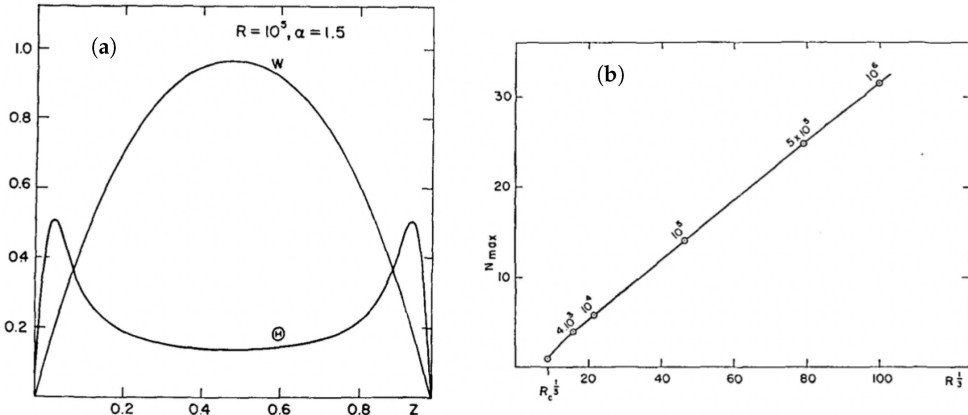

**Figure 2.** (**a**) $7.85 \times 10^{-3} w$ and $2.23\,\theta$ for $Ra = 10^5$ and $\pi\alpha = 5$. (**b**) The heat flux $Nu$ as a function of $Ra^{1/3}$ from three estimates: $N_{\max}$ is the maximum value of $Nu$ (over different $\pi\alpha$) for each given $Ra$. $N_{RS}$ is the flux predicted by the relative stability criterion, and $N_S$ is the heat transported by stable single-$\alpha$ solutions. The $N_{\max}$ coefficient given in the abstract is $C = 0.1153$. Figures 5 and 12 from Ref. [2].

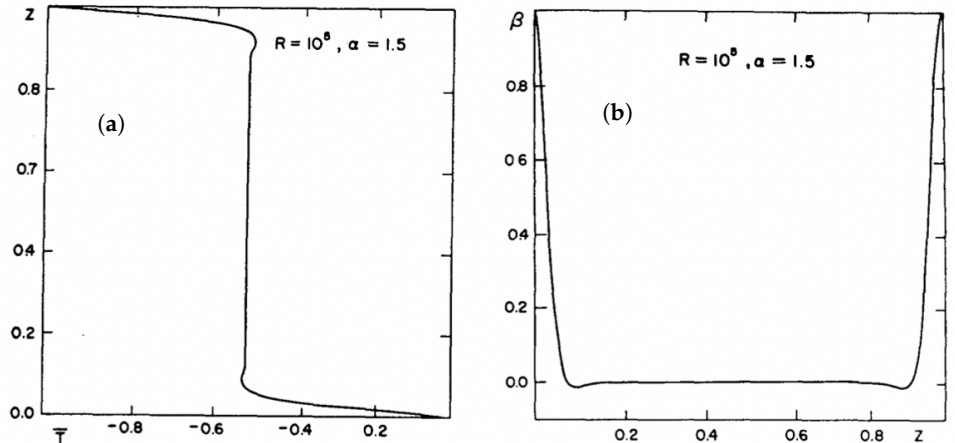

**Figure 3.** h-slip with $Ra = 10^5$ and $\alpha = 1.5$. (**a**) Mean temperature $\overline{T}(z)$. (**b**) Mean gradient $\beta(z)$. $\beta(z)$ has been normalized by the total heat transport $Nu = 13.82$. Figure 7 from Ref. [1].

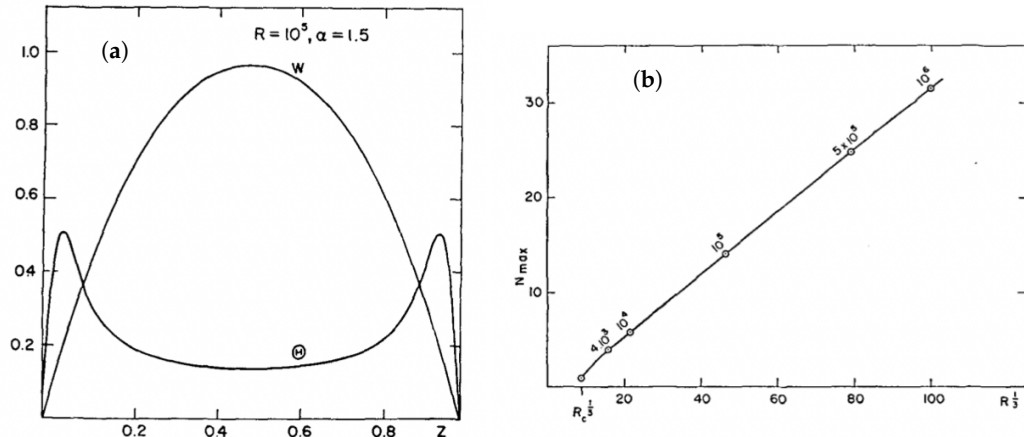

**Figure 4.** (**a**) $4.33 \times 10^{-3} w$ and $9.42\,\theta$ for $Ra = 10^5$ and $\alpha = 1.5$. (**b**) Maximum total heat flux $Nu_{\max}$ as a function of $Ra^{1/3}$. Figures 6 and 16 from Ref. [1].

**The overall message** is that, even for these early calculations, to obtain physical convective heat fluxes, rigid (no-slip) boundary conditions were required. This influenced



my decision [10] to skip free-slip boundaries and proceed straight to the task of simulating thermal convection with no-slip boundaries.

**Funding:** This research received no external funding.

**Institutional Review Board Statement:** Not applicable.

**Informed Consent Statement:** Not applicable.

**Data Availability Statement:** Not applicable.

**Conflicts of Interest:** The authors declare no conflict of interest.

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
