# Peer review of "Review of Jackson Herring’s Early Work on Thermal Convection"

_atmosphere, doi:10.3390/atmos14060907_

Round 1
Reviewer 1 Report
The paper presents a nice overview of Herring's contributions to thermal
convection and compares with our present knowledge. At the core is the
scaling of the Nusselt number with the Rayleigh number. This paper should
be published, subject to considering my comments below. I encourage
the editors to decide about the frequent use of incomplete sentences
(see details below).
line 2: "might seem primitive" doesn't seem suitable wording. Wouldn't
"might not be very impressive" be better?
line 10: why should there be a slash in "free-slip/boundary". Shouldn't
one omit the slash? In line 11, the slash might be slightly better
motivated, but even here is seems just fine to omit the slash. The
differences is that in the former, the piece "free-" applies to both
"slip" and "boundary", but this doesn't read well.
line 15: the last sentence of the abstract ("underlying message...")
suggests that it summarizes what was said above, but the statement
"physical boundaries are necessary" seems like an additional statement
that could be made clearer. What is meant here is just the no-slip
condition. By contrast, one could also have talked about a completely
boundary-free (periodic) treatment, which, I thought, would be needed
to reproduce the 1/2 scaling. A comment on this could be useful.
line 19: I'm not sure ", or bounds," makes sense and would omit it.
Eq.(1) ends with 2 commas.
line 24 (and on several further occasions) incomplete sentences are
used to immitate a more concise style. I'm not sure this is good. For
example in line 24, one could just replace "With no" by "None presented"
and the sentence would be a complete one. But there are many subsequent
sentences that are incomplete, so I'd just ask the editors to decide
whether this is acceptable.
In Eq.(3), I'd omit the 2 pairs of parentheses in the double-curl
constructs. I feel that writing \nabla\times\nabla\times without ()
is perfectly fine. The inner ones on v.nablev are ok.
2 lines above Eq.(5), "its" in "its vertical velocity" would better be
replaced by "the".
Eq.(5), it says psi(r,t), but it should just be psi(z,t).
lines 52-56, I'd combine the 2 paragraphs and try to make the sentences
more readable.
In line 62, the reference to section 5 seems to refer to Ref.[1].
If so, I'd write "section 5 of Ref.[1]."
line 63: I -> 1.
Unlike Figs.2-4, Fig.1 includes the caption from the original paper.
This part could be removed from better uniformity. Also, the caption
should say from which paper the figure is taken.
line 71, wasn't the combination of sinh ad cosh functions already
introduced in Chandrasekhar 1961?
In the caption to Fig.4, I'd add a space "\," between 5.16 and \theta.
Reviewer 2 Report
Report on: "Review of Jackson Herring’s early work on thermal convection", by: Robert M. Kerr.
As its title indicate the paper is a review of 3 of Herring's highly cited papers on thermal convection. Here are the numbers of (Google) citations for the 3 references: [1]: 215; [2]: 122; [3]: 24. As it is the review is complete and well written and I accept it, provided that the author considers the following points. I do not need to see the revised manuscript.
The results of reference [1] and [2] are well described in the review, but about reference [3] (the 1966 paper) all that is said is that:
-(a) " The third paper was an attempt to pre-condition the iterative procedure to get higher Rayleigh number results and only got the free-slip scaling" (in the abstract)
and
-(b) " The 1966 paper is not discussed in any detail because the 1963 scaling results are repeated" (in the introduction).
However, in the discussion section there are 2 sentences about the 1963 paper:
-(c) "The third paper [3] was an attempt to pre-condition and accelerate the iteration towards a steady-state in order to determine the scaling laws at much higher Rayleigh numbers. Perhaps to see if the Nu ∼ Ra1/2 ultimate scaling [6] could be found. The pre-conditioning was only applied to the free-slip geometry and reproduced the earlier free-slip results [1]."
Major point:
The author should ament the introduction sentence "The 1966 paper is not discussed in any detail because the 1963 scaling results are repeated" because (a) the paper is discussed in the conclusion and (b) this is, after all, written in memory of Herring and I believe the comments on his work should remain positive.
Minor point:
At the end of the paper it is stated that: "Which influenced my decision to skip free-slip boundaries and get straight to the task of simulating thermal convection with rigid, no-slip boundaries". It is not clear to me who's decision is mentioned: Herring's or the reviewer? Please be more precise.
Reviewer 3 Report
This is an interesting look back on the papers of Herring, with the a take-away message that even then it was realized that rigid, no-slip boundary conditions were required in order to get physical heat fluxes. I have listed some comments below mainly to improve the clarity of the manuscript which is recommended for publication in this special issue.
Page 1, line 4: insert "paper"after “first” since other things are being enumerated)
Page 1, line 5: Change a period to a colon: "... and three variables: The single mode's …"
Page 1, line 10: insert "are" after “that”
Page 1, Equation 1: Kraichnan included a logarithmic term in Ra. If that is not shown it should probably at least be mentioned.
Page 2, equation 5: “with \Beta(z)=…” implies there is a \Beta already used. It appears lower down in equation 9 so maybe this should be shifted to line 44.
Page 3, line 62: For clarity, it would be better to write “In section 5 of reference [1]”
Page 3, Figure 1: since this is copied from Herring, there should be a “After Herring [1] or something similar in the caption.
Page 4, Figure 2: same comment as above.
Page 4, line 71: Add “of ref [2]” after “appendix”
Page 4, line 77: “replace period with a comma
Page 4, line 78 Replace “Hoping” with “hoping”
Page 4, line 78: Replace the period after “us” with another comma and then replace “An” with “an”
Page 5, Figure 3: Add “After Herring [2]” in the caption
Page 5, line 83: replace period after “numbers” with a comma and replace “Perhaps” with “perhaps”
Page 5, line 100: Replace period after “required” with a comma and then replace “Which” with which”
Author Response
Report of Reviewer 3
This is an interesting look back on the papers of Herring, with the a take-away message that even then it was realized that rigid, no-slip boundary conditions were required in order to get physical heat fluxes. I have listed some comments below mainly to improve the clarity of the manuscript which is recommended for publication in this special issue.
Page 1, line 4: insert "paper"after “first” since other things are being enumerated)
Changed to “In the first two papers, 1963 and 1964, the presented results used’’
Page 1, line 5: Change a period to a colon: "... and three variables: The single mode's …"
Change: no colon, but the next line reads “The single mode's variables are its vertical velocity profile …”
Page 1, line 10: insert "are" after “that”
Fixed
Page 1, Equation 1: Kraichnan included a logarithmic term in Ra. If that is not shown it should probably at least be mentioned.
Fixed
Page 2, equation 5: “with \Beta(z)=…” implies there is a \Beta already used. It appears lower down in equation 9 so maybe this should be shifted to line 44.
Fixed the equation to put definition of beta first.
Page 3, line 62: For clarity, it would be better to write “In section 5 of reference [1]”
Fixed per another reviewer.
Page 3, Figure 1: since this is copied from Herring, there should be a “After Herring [1] or something similar in the caption.
Fixed following advice of another reviewer
Page 4, Figure 2: same comment as above.
Fixed following advice of another reviewer
Page 4, line 71: Add “of ref [2]” after “appendix”
Fixed per another reviewer.
Page 4, line 77: “replace period with a comma Page 4, line 78 Replace “Hoping” with “hoping”
Fixed
Page 4, line 78: Replace the period after “us” with another comma and then replace “An” with “an”
Fixed
Page 5, Figure 3: Add “After Herring [2]” in the caption
Fixed following advice of another reviewer
Page 5, line 83: replace period after “numbers” with a comma and replace “Perhaps” with “perhaps”
There is now a more complete discussion of what I interpret the purpose of the 3rd paper was.
Page 5, line 100: Replace period after “required” with a comma and then replace “Which” with which”
These are two separate thoughts. To make this clearer I have replaced “Which” with “This”.
Reviewer 4 Report
It is a review paper about the previous work from Herrington in the 60s. I made small comments/suggestions in order to make the paper more clearly.

fine
